Model for predicting drug resistance based on the clinical profile of tuberculosis patients using machine learning techniques

Falcao Igor Wenner Silva 1 igorufpa2013.4@gmail.com
http://orcid.org/0000-0002-5971-3668 Cardoso Diego Lisboa 1
http://orcid.org/0000-0002-0410-5032 Coutinho dos Santos Santos Albert Einstein 1
Paixao Erminio 1
Costa Fernando Augusto R. 2
Figueiredo Karla 3
Carneiro Saul 4
Seruffo Marcos César da Rocha 1
1 Institute of Technology, Federal University of Para , Belém, PA , Brazil
2 Center for Higher Amazon Studies, Federal University of Para , Belém, PA , Brazil
3 Computer Science, State University of Rio de Janeiro , Rio de Janeiro, RJ , Brazil
4 João de Barros Barreto University Hospital, Federal University of Para , Belém, PA , Brazil
Iammarino Martina
Electronic publication date: 2024 Oct 14
Publication date: 2024
Volume: 10
Electronic Location ID: e2246
Received 2024 May 16; Accepted 2024 Jul 17
Copyright: © 2024 Falcão et al.
Copyright year: 2024
Copyright holder: Falcão et al.
License: This is an open access article distributed under the terms of the Creative Commons Attribution License, which permits unrestricted use, distribution, reproduction and adaptation in any medium and for any purpose provided that it is properly attributed. For attribution, the original author(s), title, publication source (PeerJ Computer Science) and either DOI or URL of the article must be cited.
License URL: https://creativecommons.org/licenses/by/4.0/

Keywords: Tuberculosis, Machine learning, Drug resistance, Anti-tuberculosis

Funding: This work was carried out without any financial support.

==============================
Tuberculosis (TB) is a disease caused by the bacterium Mycobacterium tuberculosis and despite effective treatments, still affects millions of people worldwide. The advent of new treatments has not eliminated the significant challenge of TB drug resistance. Repeated and inadequate exposure to drugs has led to the development of strains of the bacteria that are resistant to conventional treatments, making the eradication of the disease even more complex. In this context, it is essential to seek more effective approaches to fighting TB. This article proposes a model for predicting drug resistance based on the clinical profile of TB patients, using machine learning techniques. The model aims to optimize the work of health professionals directly involved with tuberculosis patients, driving the creation of new containment strategies and preventive measures, as it specifies the clinical data that has the greatest impact and identifies the individuals with the greatest predisposition to develop resistance to anti-tuberculosis drugs. The results obtained show, in one of the scenarios, a probability of development of 70% and an accuracy of 84.65% for predicting drug resistance.

Introduction

Tuberculosis (TB) is an infectious disease caused by the bacterium Mycobacterium tuberculosis. This disease directly affects the lungs, but can also affect other parts of the body. TB is one of the main causes of death worldwide, especially in low-and middle-income countries. It remains a major global health problem, with an estimated 10.6 million cases and 1.6 million deaths worldwide in 2021 (Chakaya et al., 2022). The disease is mainly transmitted by airborne droplets through coughing or sneezing, but transmission can also occur through close contact with infected individuals or contact with infected body fluids (Das et al., 2022).

In developing countries, tuberculosis has been a serious health problem, with approximately 282,000 new cases reported in recent years and a mortality rate of 2.4 per 100,000 inhabitants (Orjuela-Cañón et al., 2022). In Brazil, the TB mortality coefficient declined year on year, albeit slowly, for approximately two decades, until this trend was reversed in 2021, when 5,072 deaths were recorded, giving a coefficient of 2.38 TB deaths per 100,000 inhabitants. The description of this scenario seems to put the achievement of the global targets and commitments for the elimination of TB out of perspective (do Amazonas, 2024). It is important to note that tuberculosis is a slow-growing disease, with a replication cycle every 24 h, which contributes to the manifestation of subacute symptoms (Orjuela-Cañón et al., 2022).

In TB patients, the risk of transmission to healthy individuals also depends on risk factors associated with the patient’s other clinical conditions, such as social conditions, the occurrence of comorbidities and other related aspects (Ashna et al., 2018). In this context, factors such as AIDS, economic situation, homelessness and alcohol abuse, as well as smoking, have a direct impact on the development of the disease (Amiri, Siami & Khaledi, 2018). Awareness of these factors is fundamental, as they play a crucial role in preventing and combating this disease.

Early detection of TB is of paramount importance for effective control of the disease. Diagnostic methods, such as the tuberculin test, are widely used in developing countries to detect latent TB (Gill et al., 2022). However, these methods face challenges due to the lack of resources and adequate infrastructure in many regions, which can lead to the occurrence of drug resistance that can arise during the therapeutic course. In some cases, the bacteria can develop resistance to the drugs used in treatment, which compromises the effectiveness of the therapeutic regimen and increases the severity of the disease. This situation is a serious threat to public health and calls for innovative approaches, such as the use of computational intelligence, to study and understand drug resistance patterns in order to develop more effective and personalized treatment strategies (Dheda et al., 2014).

There are two forms of drug resistance to tuberculosis: multidrug-resistant (MDR-TB) and extensively drug-resistant (XDR-TB). In MDR-TB, the patient’s organism becomes resistant to the drugs Isoniazid and Rifampicin, while in XDR-TB there is resistance to isoniazid, rifampicin, a fluoroquinolone and a second-line injectable (Amikacin, capreomycin and kanamycin) or isoniazid, rifampicin, a fluoroquinolone and bedaquiline or linezolid (Cheepsattayakorn, 2013). Treating tuberculosis is financially costly, although severe cases of the disease are uncommon. In addition, there are other conditions that can manifest after recovery from tuberculosis, such as hearing loss, depression or hepatitis, in some cases (Duwairi & Melhem, 2023).

Resistance to the drugs used to treat tuberculosis represents a considerable obstacle in the fight against this disease. Inappropriate use of drugs and premature discontinuation of treatment can lead to the emergence of bacterial strains that are immune to the drugs (Nadgir et al., 2023). Another worrying factor is the relationship between drug resistance and opportunistic diseases. Patients with compromised immune systems become more susceptible to secondary infections, which increases their severity and makes the recovery process even more difficult. This has made it essential to effectively address drug resistance (BRASIL, 2018).

Chemotherapy for TB treatment, especially in cases of drug resistance, is complex and requires careful drug management and control strategies. In many countries, TB treatment is often based on empirical diagnoses, which can lead to additional challenges in the effective management of the disease (Rabahi et al., 2017). Finally, TB represents a significant global public health challenge, requiring a multidisciplinary approach that includes improvements in public health, education, diagnosis and treatment. International collaboration and a commitment to research are essential to eradicate this devastating disease or at least mitigate its damaging effects.

The application of advanced techniques, such as machine learning, has promising potential in tackling drug resistance to tuberculosis. These approaches can be used to carry out a thorough analysis of information on patients undergoing treatment, taking into account various clinical, physical and social aspects, as well as the results of laboratory tests. By exploring this data in an integrated way using data science techniques, which is one of the aims of this study, it is possible to gain valuable insights that contribute to a deeper understanding of drug resistance, helping to make more precise and personalized clinical decisions.

Given the issues presented, the contribution of this work stands out for the creation of a prediction model capable of anticipating the clinical outcome of patients in relation to TB drug resistance. The model not only treats the data sensitively, but covers a variety of functionalities, from data analysis, processing and transformation to grouping individuals with similar characteristics, in addition to a data set already consolidated in the literature, used by Yamaguti et al. (2020, 2018) and Crepaldi et al. (2019). Notably, the review of the state of the art reveals a substantial gap in the specific field of drug resistance, where such comprehensive models are scarce and with a limited domain for machine learning, as shown by Kouchaki et al. (2019), Liao et al. (2023), Deelder et al. (2019) and Kuang et al. (2022).

The literature, to date, stands out for the lack of a comprehensive approach that encompasses everything from data collection and processing to personalized prediction, which is aimed at obtaining insights into which characteristics most influence or who is most likely to obtain a positive picture for drug resistance. Thus, this work fills a gap identified in the existing scientific literature, offering an advanced perspective that, through a robust model based on real data from tuberculosis patients and, by using machine learning as a mode of prediction, can act to tackle this public health problem.

This article is organized as follows: “Related Works” provides a comprehensive bibliographic survey of related research. “System Model” describes the methodology and structure of the patient profile analysis system. “Results and Discussion” presents the results obtained and an in-depth analysis of the findings. Finally, “Conclusion” ummarizes the key points and presents the final considerations of the study.

Related works

Multidrug-resistant tuberculosis (MDR-TB) is a form of the disease that is resistant to at least two of the most effective drugs currently available for treating tuberculosis, rifampicin and isoniazid (Chakaya et al., 2021). This MDR-TB problem, despite intense efforts to combat it, continues to rise globally, emerging as a growing public health concern. Various approaches are being adopted to tackle this issue, such as the implementation of longer treatments, the use of combination therapies and the research and development of new drug formulations.

Drug resistance in tuberculosis is a complex challenge, requiring multifaceted strategies and continuous innovation to contain it. In addition to the aforementioned measures, education and adequate support are fundamental for adherence to treatment; machine learning and data science models are used for this. Thus, the use of artificial intelligence (AI) during treatment is emerging as a proposal with great potential to benefit patients and the rigorous monitoring of the progress of drug resistance. It is important to note that several studies are actively involved in this area, seeking to further improve these approaches.

Hussain & Junejo (2019) discuss drug succession in patients being treated for tuberculosis. The problem was approached with classification using three machine learning algorithms (Random Forest, support vector machine and neural networks), evaluating 4,213 patients, 64.37% of whom completed treatment. The models showed an improvement of more than 12% in accuracy compared to the baseline, offering insights to improve TB programs. This methodology can help TB program teams manage human resources more effectively, potentially saving more lives.

In Deelder et al. (2019) tuberculosis and drug resistance are investigated in the face of the drug regimen offered free of charge. This study used whole genome sequencing on 16,688 isolates of M. tuberculosis and drug sensitivity tests for 14 anti-tuberculosis drugs. Machine learning models including classification trees and gradient-boosted trees were applied to predict drug resistance and identify new associated mutations. The models were adjusted for each drug and evaluated on performance metrics. However, the work is limited by not covering clinical and epidemiological data, which could enrich the analysis and improve the accuracy of drug resistance predictions.

The growing proportion of drug-resistant tuberculosis represents a serious public health problem. In this context, Liang et al. (2022) have developed artificial intelligence (AI) methods, with a focus on deep learning, to detect pulmonary tuberculosis, distinguish the infection from other lung diseases and identify tuberculosis drug resistance in medical images and genetic data. The aim is to help doctors choose the right treatment in the early stages of the disease, in order to maximize the impact of AI in this field. However, it is important to note that the study was unable to identify resistance-related mutations and considered a limited number of variables.

Drug resistance in tuberculosis is observed in different forms of the disease, as discussed by Duwairi & Melhem (2023); in this study, deep learning models were used to identify tuberculosis based on computed tomography images. These models use the VGG19 and ResNet neural networks to extract relevant features from the images. The most effective model for classifying multidrug resistance achieved an accuracy of 74.13% and an area under the curve (AUC) of 64.2%. Although the study showed significant results, there are notable limitations, such as the scarcity of data in the dataset and the limited consideration of performance metrics in the evaluation of the models.

Identifying tuberculosis drug resistance is crucial to combating antibiotic resistance. The Kouchaki et al. (2019) study used machine learning to predict resistance in MDR-TB (multi-drug-resistant TB) samples from 16 countries. The results showed significant improvements compared to conventional tests, especially for drugs such as pyrazinamide, ciprofloxacin and ofloxacin. Logistic regression and gradient tree augmentation stood out. However, it is considering a limited data set that can lead to overfitting models, achieving optimistic but not generalizable results.

Despite the growing number of studies evaluating tools for analyzing drug resistance profiles using data, as exemplified by references such as Yamaguti et al. (2020, 2018) and Crepaldi et al. (2019), the concepts and applications of these approaches still lack a more comprehensive discussion in the most robust environment. Notably, the potential of machine learning (ML) in this context has perhaps not yet reached its full utilization in clinical applications. The main objectives of this study are: (I) To develop a model for predicting drug resistance based on the clinical profile of tuberculosis patients using ML techniques. This involves using ML techniques to analyze patient data and identify patterns that can indicate resistance;

(II) Explore and analyze the drug resistance profile in TB using ML models. This includes examining how different factors contribute to resistance and understanding the overall resistance landscape in the patient population.; and

(III) To specify which characteristics have the greatest impact on patients’ involvement with drug resistance. This involves pinpointing the most influential factors that contribute to resistance, helping to better target interventions and treatments.

The implementation of ML-based prediction models for drug resistance represents a promising revolution in healthcare. These models have the potential to improve the capacity for early diagnosis of drug resistance, allowing for a more personalized and efficient approach to the treatment of tuberculosis. By incorporating advanced algorithms, the model can identify complex patterns in the data, making it possible to accurately predict certain drugs. This not only speeds up the resistance prediction process, but also optimizes the choice of drugs, potentially reducing treatment time and adverse effects associated with the use of inappropriate drugs. Furthermore, the successful implementation of ML models in the evaluation of resistance profiles can contribute to a more assertive approach to controlling the spread of tuberculosis.

System model

In this article, a prediction model is proposed to analyze the profile of patients being treated for tuberculosis in relation to the development of drug resistance. Machine learning techniques were incorporated into the model to classify the patients most likely to develop resistance, as well as the factors related to this condition. The model incorporates four stages (Fig. 1): Pre-processing, Clustering, Segmentation and Learning. In addition, it integrates a dataset covering a variety of information.

Figure 1 The figure illustrates the workflow of the prediction model according to the proposed architecture, which has four execution steps (Preprocessing, Clustering, Segmentation, and Learning).

Execution is limited to the training flow, represented by a solid line. The figure was created using the Draw.io tool from Google (https://www.drawio.com/doc/faq/usage-terms). All icons used in the figure are free icons available in Draw.io and can be used under the Creative Commons license (CC BY 4.0).

The model has a high adaptive capacity, which suggests that it can be applied to other pathologies, making it a versatile tool for health analysis and intervention. As can be seen in Fig. 1, the model is able to quantify categorical data by assigning numerical values to the categories, resulting in an ideal linear regression equation for the transformed variables.

Figure 1 illustrates the workflow of the prediction model according to the proposed architecture. Execution is limited to the training flow, represented by a continuous line. In the data processing stage, patient records are pre-processed and fed directly into the clustering model. This process aims to identify affected groups with similar characteristics. The desirable number of clusters identified using the Elbow method (Thorndike, 1953) was three, which was determined after testing the model with various cluster quantities to validate the quantity suggested by the method. This process involved evaluating parameters such as the total within-cluster sum of squares (WSS) as a function of the number of clusters. This technique is often adopted because of its visual and intuitive approach to identifying the balance point.

Each Cluster in Fig. 1 is characterized by segmentation, which divides the data set into homogeneous groups based on shared characteristics. The definition of the Cluster is followed by the calculation of the probability of developing drug resistance, which is carried out in the learning stage of the implemented models. The results not only demonstrate the predictive capacity of the model, but also reveal the probability distribution for resistance to specific drugs during treatment. The insights indicate a possible direct relationship between the increase in cases of drug resistance and the clinical condition of the patients.

Dataset

This study analyzed data from 103,846 records of tuberculosis patients in the state of São Paulo, obtained from 2006 to 2016 through TBWEB, an information system used for monitoring tuberculosis, available in a public repository (https://figshare.com/articles/dataset/tuberculosis-data-06-16_csv/8066663) public. The dataset includes clinical, social and laboratory information on patients, including follow-up details such as drug resistance to the main drugs administered by the Unified Health System (SUS, in Portuguese), such as rifampicin, isoniazid, ethambutol and the alternative scheme. It is worth noting that the alternative regimen can vary, incorporating drugs in specific combinations and for specific periods (Biagioni, Cavicchioli & Massabni, 2022).

From the aforementioned set, focusing on the resistance attribute (indicating patients’ drug resistance), a subset was extracted containing 1,536 records, which were transformed and showed no outliers. Of these records, 50% correspond to individuals with confirmed drug resistance, while the other 50% were randomly selected from the original set. The selection was carried out using the Random State method with a specific seed, ensuring control of sample variability for this type of problem, which makes it possible to obtain the same sequence of random numbers when running the model again.

Data preprocessing

The data set, in its original form, consisted of an extensive amount of attributes, mainly categorical data, covering both nominal and ordinal variables. During pre-processing, the data was discretized and subjected to Chi-square analysis (Plackett, 1983) to test the independence of categorical variables and determine significant associations between them. Using this method, it was possible to verify the association between the attributes in the data set and the variable of interest by comparing the observed and expected frequencies. This method was chosen because of the initial characteristics of the data set, which had a variety of categories, as well as problems such as missing values, inconsistencies and duplicates. These issues required a comprehensive approach to data cleaning and preprocessing to ensure the accuracy and reliability of subsequent analyses. By addressing these data quality issues systematically, intended to enhance the robustness of our findings and minimize the potential impact of erroneous or misleading data on the results.

During the statistical analysis, we identified the attributes of the data set that were significantly associated with the occurrence of drug resistance in individuals, which is the variable of interest in the model. A significance level ( α) of 0.05 was adopted, which means that there is a 5% probability of rejecting the null hypothesis when it is true. To evaluate this association, we used the library Statistical Functions (scipy.stats; https://docs.scipy.org/doc/scipy/reference/stats.html) in Python. After the process, 29 attributes were selected (as shown in Table 1) from a set of variables that, according to the Chi-square test, have a statistically significant association with the manifestation of drug resistance, allowing us to focus on the most informative attributes for the model.

Table 1 The table covers a variety of features related to the profile of patients being treated for tuberculosis.

Each column represents a single feature, while the rows provide different categories associated with each feature. It should be noted that the table depicts the input data relationship for a model, displaying specific values for each feature. Notably, the output label, represented by the “Status_Resistance” attribute, is categorical in nature and is divided into two distinct classes.

ID	Attribute	Description	Value	
1	Age group	Age group	Thirteen categories	
2	Gender	Gender	Female or male	
3	Education level	Scholarity	Seven categories	
4	Type of occupation	Type of occupation	Eight categories	
5	Current status	Current state	Sixteen categories	
6	Type of case	Type of case	Six categories	
7	Clinical form1	Clinical form	Fifteen categories	
8	Classification	Classification	Five categories	
9	Method of discovery	Case-finding method	Nine categories	
10	Bacteriological exam	Smear microscopy	Five categories	
11	Other bacterial exam	Smear microscopy with another material	Four categories	
12	Sputum culture	Sputum culture	Five categories	
13	XRay	Chest X-Ray	Six categories	
14	Necropsy	Necropsy	Four categories	
15	HIV	HIV	Five categories	
16	Sensitivity testing	Sensibility test	No information, Yes or No	
17	AIDS	AIDS	Yes, No	
18	Diabetes	Diabetes	Ye, No	
19	Alcoholism	Alcoholism	Yes, No	
20	Mental illness	Mental disease	Yes, No	
21	DrugAddiction	Drug addiction	Yes, No	
22	Smoking	Smoking	Yes, No	
23	Treatment institution	Institution type	Nine categories	
24	Initial scheme	Initial treatment scheme	Six categories	
25	Current scheme	Current scheme	Six categories	
26	Reason for scheme change	Scheme change reason	Four categories	
27	Treatment type	Treatment type	No information, observed, non-observed	
28	Histopathology	Histopathological	Four categories	
29	Status_resistencia	Resistance	Yes, No	

Table 1 lists the model’s input data and their respective values, including the output label represented by the “Status_Resistance” attribute. This output label (Categorical) of the model is distributed between two classes. In cases of missing information, these records are disregarded during pre-processing.

Specification of clinically affected groups

The specification of groups initially involves running algorithms to identify clusters and determine the optimum number. Clustering models were run, including the K-Modes algorithm (Guha, Rastogi & Shim, 2000), Rock Clustering (Guha, Rastogi & Shim, 2000) and DBSCAN (Khan et al., 2014), with the aim of identifying groups with similar characteristics and predicting the best probability distribution for each group. During the experiment, K-Modes stood out as the best model for clustering. Its specialization in dealing with categorical data and the efficient identification of the most frequent values in each cluster demonstrated significant effectiveness in clustering categorical data in relation to the input data set.

The experiment was conducted comprehensively to validate the effectiveness of the KMODES algorithm in comparison with other algorithms. This becomes crucial in the process of identifying patients with similar characteristics, and is essential for personalizing therapeutic approaches, adapting treatments and identifying patterns of behavior, symptoms and shared risk factors. The parameters (n_clusters = 3, init = ‘Huang’, n_init = 20, max_iter = 100, random_state = 2) were carefully selected to optimize the performance of KMODES. This choice was based on extensive testing with various combinations, ensuring that the algorithm was adequately adjusted to the specific characteristics of the data set. Table 2 presents examples of tests performed.

Table 2 Evaluation of kmodes hyperparameter tuning.

Initialization method	Number of initializations	Max. number of iterations	Cost	
Huang (1997)	20	50	4,287	
Huang (1997)	20	100	4,287	
Huang (1997)	10	20	4,291	
Random	20	100	4,308	
Random	10	20	4,327	
Cao et al. (2012)	10	50	4,402	

The initialization method determines the initial choice of centroids, which can significantly affect the quality of the resulting clusters. The number of initializations specifies how many times the algorithm will be run with different starting points. This is important for increasing the likelihood of finding the best solution, minimizing the impact of randomness in centroid initialization. The maximum number of iterations limits how many times the algorithm recalculates the centroids and reassigns instances to clusters. This step aims to avoid infinite loops and ensure that the algorithm finishes in a reasonable time, even if it does not completely converge. The total clustering cost is the sum of the distances of the instances to their respective centroids in the formed clusters. This metric is used to evaluate the quality of the clusters, with a lower cost generally indicating a better fit of the clusters to the data.

In KModes, which is used for categorical data, the “distance” between two instances is measured as the Hamming distance. The Hamming distance is the count of the number of attributes that differ between two instances. Thus, the total cost is calculated as follows: Assignment of centroids: Each instance is assigned to the nearest centroid, where proximity is measured by the smallest Hamming distance.

Sum of distances: For each instance, the distance (number of differing attributes) to its assigned centroid is calculated. The total cost is the sum of these distances for all instances.

The clustering generated by KModes aims to identify individuals with clinical characteristics similar to tuberculosis, making it easier to visualize shared characteristics. Efficient categorization seeks to provide valuable insights for implementing strategies against drug resistance. KModes’ ability to handle categorical data makes it a suitable choice for analyzing specific clinical aspects of tuberculosis, contributing to informed decisions and improving personalized therapeutic approaches.

Metrics and machine learning models

After categorization, logistic regression was used in the final stage of the model to assess the probability of developing drug resistance to tuberculosis treatment in each group. The analysis only considered individuals with no history of drug resistance in each group. Regression was applied to each group and filtering was applied to form subgroups composed only of patients without confirmed resistance. This approach allows the conversion of log-chances (i.e., the greater the chances, the greater the log of the chances and vice versa) into probabilities, making logistic regression effective in classifying instances.

The ML algorithms applied in this study are based on Adaboost, Random Forest (RF), support vector machine (SVM), XGBoost and logistic regression and were applied to identify cases of tuberculosis drug resistance in patient clinical follow-up data. These models have been widely and successfully used for this type of task. To implement the models, the scripts used Python 3.7.6 (https://www.python.org/) and the scikit-learn library (https://scikit-learn.org/stable/), numpy (https://numpy.org/), pandas (https://pandas.pydata.org/), matplotlib (https://matplotlib.org/) and imblearn (https://imbalanced-learn.org/stable/). Machine learning models tend to learn little from minority class data. Therefore, data balancing is fundamental when training predictive models.

Evaluation metrics are crucial for quantifying the performance and effectiveness of machine learning models, playing an essential role in the prediction stage of the model proposed in this work. Among the various metrics discussed are classification metrics such as accuracy, specificity and F1-score, which assess the model’s ability to correctly classify positive and negative instances. In addition, metrics such as the ROC (receiver operating characteristic) curve and the area under the curve (AUC) are used to assess performance in binary classification problems.

This study was submitted to Plataforma Brasil and has the number CAAE: 70947317.3.0000.5440. This study was approved by CEP (Comitê de Ética em Pesquisa, i.e., Committee of Research Ethics)/CONEP (Comissão Nacional de Ética em Pesquisa i.e., National Research Ethics Commission) of the Brazilian Government.

Results and discussion

The study focused on an initial understanding of the data related to tuberculosis, especially drug resistance, which is a mechanism by which patients often develop an inability to respond to drugs. Although the treatment regimen is constantly being developed, there is discussion about the possibility of the disease becoming untreatable in some cases. Therefore, the development of new therapeutic options to combat emerging drug resistance is essential (Ferreira Neto, Oliveira & Pimenta, 2020).

In the exploratory data analysis (EDA) carried out to identify patterns, trends and resistance before applying specific learning techniques, the relationships between various diseases were explored, including diabetes, alcoholism, drug addiction, HIV and smoking. An important highlight was the finding that patients with co-infection of TB and HIV, which is a determining factor for the increase in the mortality rate, obtained up to 80% after confirmed diagnosis, as evidenced in studies relevant to the area (Cheepsattayakorn, 2013; McDowell & Pai, 2016).

The techniques applied resulted in the identification of three clusters (Cluster A, Cluster B and Cluster C), representing different resistance profiles in tuberculosis patients. The analysis revealed specific characteristics associated with resistance, highlighting the opportunistic diseases that coexist in the individuals in each cluster, as seen in Fig. 2. This approach contributes significantly to the identification of risk factors and the development of more effective strategies for the management of drug-resistant tuberculosis.

Figure 2 Three distinct clusters regarding opportunistic comorbidities and drug resistance in tuberculosis patients.

In Cluster A, smoking and alcoholism are associated with high resistance to antituberculosis drugs. In Cluster B, only alcoholism stands out as a risk factor. In Cluster 3, HIV and drug addiction show a strong association with drug resistance, increasing the risk for patients with HIV/TB coinfection. These results underscore the importance of considering multiple factors in tuberculosis treatment. The figure was created using the Python programming language with the Plotly library (https://plotly.com/python/). All icons used in the figure are free icons available under the Creative Commons license (CC BY 4.0).

Figure 2 presents the percentage of patients in each cluster who have specific comorbidities. This analysis is important for demonstrating the relationship between the comorbidities presented and drug resistance. In Cluster A, with 857 patients, there is a high prevalence of resistance associated with smoking and alcoholism. This means that, for patients being treated for tuberculosis, there is a higher incidence of resistance to anti-tuberculosis drugs due to smoking and excessive alcohol consumption. In Cluster B, only alcoholism stands out, while increased resistance among individuals is perceived. This scenario indicates that the presence of alcoholism is associated with increased resistance to anti-tuberculosis drugs.

In Cluster 3 (Fig. 2), the analysis reveals that the presence of HIV and drug addiction are strongly associated with resistance to anti-tuberculosis drugs. This indicates that TB patients who also have HIV, i.e., have HIV/TB co-infection and a history of drug addiction, face a significantly higher risk of developing drug resistance. For this reason, TB infection has been one of the most common causes of death in HIV-infected patients, accounting for 26% of deaths related to the disease, with 99% of this percentage occurring in developing countries (Ji et al., 2018).

Cluster analysis revealed distinct patterns of resistance to tuberculosis treatment in different segments. The probability distribution by cluster, illustrated in Fig. 3, offers insights into the potential impact of drug resistance. The approach adopted strengthens the ability to optimize resources and improve results in the fight against tuberculosis. It was observed that certain clusters are more prone to resistance. In this case, logistic regression not only demonstrated its effectiveness in predicting resistance, but also identified clusters with similar characteristics more efficiently when compared to the other algorithms mentioned in the study.

Figure 3 The results of probability distribution using logistic regression.

The larger the “candle”, the greater the probability dispersion within each cluster. The Figure was created using the Python programming language with the Plotly library (https://plotly.com/python/). All icons used in the figure are free icons available under the Creative Commons license (CC BY 4.0).

Figure 3 shows the results of calculating the probability distribution using logistic regression. The calculation is performed in Python using the function “predict_proba” (https://scikit-learn.org/stable/) which returns the estimated probability for each output class of the model, i.e., the values per Cluster. The larger the “candle”, the greater the dispersion of probability between the individuals in each Cluster, as can be seen in Cluster B, where a high probability of developing drug resistance was obtained. Initially, Cluster A was composed mostly of individuals affected by alcoholism and smoking, with a small number of individuals affected by HIV and drug addiction.

In Cluster A, which is also shown in Fig. 3, the average probability of acquiring resistance was 70%. In Cluster B, the average probability of developing resistance was 66.5%, and finally, Cluster C had a probability of 23.2%, where the percentage of individuals per cluster and the probability of these individuals presenting resistance are expressed, considering their clinical characteristics. The average values were calculated and shown in Table 3.

Table 3 A variety of features related to the profile of patients being treated for tuberculosis.

Each column represents a single feature, while the rows provide different categories associated with each Feature.

ID Cluster	Patients without resistance (%)	Probability average (%)	
Cluster A	29%	70%	
Cluster B	89%	66%, 5%	
Cluster C	40%	23%, 2%	

In general, the classes in the model have varying numbers of individuals, which does not directly interfere with the execution of the probabilistic model. In the experiment shown in Table 3, the highest probability of developing drug resistance was observed in Cluster A, reaching 70%. This indicates that patients belonging to this group, who do not initially have resistance, have the highest probability of developing drug resistance, considering a significance level of 0.05.

In addition to the stage of grouping and specifying the groups clinically affected by drug resistance, the proposed model includes the stage of implementing machine learning algorithms in order to provide a stage for predicting drug resistance based on the clinical profile of tuberculosis patients using learning techniques. The algorithms were tested with different combinations of parameters to determine the most effective configuration for accurate predictions, as illustrated in Table 4. The table provides detailed information on the performance metrics of each algorithm combination.

Table 4 A detailed analysis of two hyperparameters considered in the algorithms of this study (Adaboost, Random Forest, SVM, XGBoost and logistic regression).

Hyperparameters are values that define the behavior of the machine learning algorithm and directly influence its performance.

Algorithm	Hyperparameters	
Adaboost	depth = {“max depth”: [1,2,3],	
	learning rate = [0.05, 0.1, 0.2, 0.3, 0.4, 0.5, 0.6, 0.7, 0.8, 0.9, 1],	
	‘n estimators’: [50, 100, 150, 200],	
	‘min samples leaf’: [1, 2, 3, 4, 5, 6, 7, 8, 9, 10]}	
Random Forest	depth = {“max depth”: [3,4, 5, 6, 7, 8, 9, 10],	
	“min samples split”: [2, 4, 8, 12, 16],	
	‘n estimators’: [50, 100, 150],	
	‘criterion’: [gini, entropy],	
	‘max features’: [auto, 2, 3, 4, 6, 8, 10,11],	
	‘min samples leaf’: [2, 3, 4, 5, 6, 7, 8]	
XGBoost	depth = {[3, 4, 5, 6],	
	‘learning_rate’: [0.01, 0.05, 0.1, 0.2],	
	‘n_estimators’: [50, 100, 150, 200],	
	‘min_child_weight’: [1, 2, 3, 4],	
	‘subsample’: [0.8, 0.9, 1],	
	‘colsample_bytree’: [0.8, 0.9, 1]}	
Logistic Regression	{‘penalty’: [‘l1’, ‘l2’, ‘elasticnet’, ‘none’],	
	‘C’: [0.001, 0.01, 0.1, 1, 10],	
	‘solver’: [‘newton-cg’, ‘lbfgs’, ‘liblinear’, ‘sag’, ‘saga’],	
	‘max_iter’: [50, 100, 200]}	

The choice of hyperparameters, as indicated in Table 4, is a task that directly impacts the performance of the prediction model. Its definition was supported by the Grid Search method (Lerman, 1980), which performs an exhaustive search for specific combinations of hyperparameters, covering the entire predefined space, since it fully explores the set of possible values. The division of the data set for this stage was 80% for training and 20% for testing, using cross-validation (CV) with k equal to 10.

When evaluating the performance of the machine learning algorithms, metrics were selected for prediction (Accuracy, ROC-AUC and F1-score). This stage aims to identify the occurrence of drug resistance in individuals undergoing clinical treatment who had not yet received this classification. The model was developed and trained on a dataset that was carefully segmented into training and testing subsets. This proactive data curation approach ensures the model’s robustness and reliability, which is crucial for optimizing TB treatment. By analyzing various patient data, including clinical features, the model can enable early and personalized interventions. These personalized treatment plans can significantly improve patient outcomes by identifying those at higher risk and tailoring treatment strategies accordingly. The detailed results of this analysis, which highlight the model’s effectiveness and accuracy, are presented in Table 5.

Table 5 A comprehensive overview of the performance evaluation of different learning models used in this study.

Algorithm	Accuracy	ROC-AUC	F1-Score	
Adaboost	0.8285	0.854	0.835	
Random forest	0.843	0.887	0.851	
SVM	0.841	0.878	0.849	
XGBoost	0.846	0.888	0.853	
Logistic regression	0.835	0.891	0.842	

The results shown in Table 5 highlight the values achieved in the algorithms’ performance metrics during the prediction stage. It can be seen that XGBoost achieved the highest accuracy (84.65%) and the highest F1-score (85.34%) during the evaluation. On the other hand, logistic regression achieved the best results in terms of ROC-AUC (89.16%). The results show the potential of machine learning algorithms in healthcare decision-making, highlighting their superior performance in metrics for predicting drug resistance in patients during treatment.

Incorporating these models into the clinical environment, especially XGBoost, has consistently shown the best results. This was evidenced by evaluating its performance in a case consisting of 27 other models tested. Of these, only five were selected. They were chosen because they showed the most favorable results in terms of the evaluation metrics. XGBoost’s results, compared to the others, highlight its ability to deal with complexities in the data, helping to prevent overfitting and capture non-linear interactions. Its accuracy and ability to handle heterogeneous data are key to optimizing clinical processes.

Conclusion

Drug resistance in tuberculosis has been a growing challenge in recent years, jeopardizing the effective treatment of the disease. Drug resistance is a factor that can impact patients at various stages of treatment. During the COVID-19 pandemic, there has been a significant reversal in global advances against tuberculosis, undermining efforts to combat the disease. Although it is possible to identify resistance through drug sensitivity testing, the process has limitations due to external factors such as opportunistic diseases, social conditions or treatment abandonment.

To reduce the burden of these limitations, the use of the proposed model based on data pre-processing and machine learning is one of the main alternatives for dealing with resistance. This is especially important to increase the effectiveness of the treatment regimen, ensuring that patients receive drugs to which the bacteria are sensitive. Considering two classes of individuals (with and without drug resistance), the model calculated the probability of developing resistance to the drugs used in the treatment of tuberculosis, as well as providing direct evidence that HIV linked to alcoholism and drug addiction increases the probability of individuals obtaining drug resistance.

The K-modes algorithm was used for clustering and logistic regression was used to distribute the probability of resistance, with a probability of 70% for Cluster A. In the evaluation, various drug resistance prediction models were also tested, with XGBoost showing the best results, achieving an accuracy of 84.65% and an F1-score of 85.34%. For future work, we intend to consider other attributes, data on the composition of drugs and information on patients’ household contacts. It should be noted that the specification of resistance is still essentially clinical, so tools of this nature can help the whole process.

There are a few noteworthy limitations to this study.

The methodology is tailored for predicting probabilities and clustering patients based on a single disease, limiting its immediate applicability to other medical conditions.

The accuracy and robustness of the models heavily rely on the effectiveness of feature selection methods. Variability in feature relevance across diseases may impact the generalizability of the approach.

Interpretation of clusters and their meaningful clinical implications may be challenging, especially when dealing with complex, high-dimensional data.

The accuracy of predictions and clustering outcomes is contingent upon the quality, completeness, and representativeness of the medical data.

While the methodology may demonstrate efficacy within a specific healthcare setting or population, its generalizability across different healthcare systems, demographics, and geographical regions warrants further investigation and validation.

Supplemental Information

Supplemental Information 1 Dataset providing comprehensive information and insights into tuberculosis (TB).

Supplemental Information 2 Code.

This work received institutional support from the National Council for Scientific and Technological Development (CNPq) and the Pro-Rectory for Research and Postgraduate Studies at the Federal University of Para by facilitating academic research and scientific initiatives carried out by its collaborators.

Additional Information and Declarations

Competing Interests

Author Contributions

Ethics

Data Availability

The authors declare that they have no competing interests.

Igor Wenner Silva Falcao conceived and designed the experiments, performed the experiments, analyzed the data, performed the computation work, prepared figures and/or tables, authored or reviewed drafts of the article, wrote the code, and approved the final draft.

Diego Lisboa Cardoso conceived and designed the experiments, performed the experiments, analyzed the data, prepared figures and/or tables, authored or reviewed drafts of the article, and approved the final draft.

Albert Einstein Coutinho dos Santos Santos conceived and designed the experiments, performed the experiments, analyzed the data, performed the computation work, prepared figures and/or tables, authored or reviewed drafts of the article, wrote the code, and approved the final draft.

Erminio Paixao conceived and designed the experiments, analyzed the data, prepared figures and/or tables, authored or reviewed drafts of the article, and approved the final draft.

Fernando Augusto R Costa analyzed the data, prepared figures and/or tables, authored or reviewed drafts of the article, and approved the final draft.

Karla Figueiredo conceived and designed the experiments, prepared figures and/or tables, authored or reviewed drafts of the article, and approved the final draft.

Saul Carneiro performed the experiments, prepared figures and/or tables, authored or reviewed drafts of the article, and approved the final draft.

Marcos César da Rocha Seruffo analyzed the data, prepared figures and/or tables, authored or reviewed drafts of the article, and approved the final draft.

The following information was supplied relating to ethical approvals (i.e., approving body and any reference numbers):

This study was submitted to Plataforma Brasil and has the number CAAE: 70947317.3.0000.5440. This study was approved by CEP (Comitê de Ética em Pesquisa, i.e., Committee of Research Ethics)/ CONEP (Comissão Nacional de Ética em Pesquisa i.e., National Research Ethics Commission) of the Brazilian Government.

The following information was supplied regarding data availability:

The data are available at Figshare: Yamaguti, Verena (2019). Tuberculosis treatment dataset. figshare. Dataset. https://doi.org/10.6084/m9.figshare.8066663.v2.

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
