# Peer review of "Model for predicting drug resistance based on the clinical profile of tuberculosis patients using machine learning techniques"

_PeerJ Computer Science, doi:10.7717/peerj-cs.2246_

## Round 0.1 · original submission · Minor Revisions

Few changes are required to improve the work before publication.

Would be appropriate :
- better motivate the research questions defined in the work
- reread and correct writing errors
- improve the transition between one section and another
- insert a figure showing some examples of different choices on the clustering hyperparameter

Reviewer 1 ·

Basic reporting

The work has a very clear objective, and the results obtained are quite satisfactory with respect to the field of application, considering the small number of instances.

The article uses clear and technically correct text.
There are some typos, such as on page 5 in the Data Preprocessing section, it is written, "we used the library Statistical Functions (scipy.stats2) em Python", but it should be 'in Python'.

The literature is described sufficiently and clearly, but I suggest inserting the quotes in brackets to facilitate reading. Furthermore, the link on page 5 which refers to the dataset is incorrect as it is missing an underscore.

The figures appear to be clear for the purpose and well explained, however I suggest inserting a figure as a result of the application of the Elbow method to verify that 3 is the suitable number as the number of clusters.

Experimental design

The objective of the work is clear, but it would help to understand the drafting of the Research Questions to understand the steps that led to making specific design choices and the related results.

The methods used have been sufficiently described, and additional material is suitable for replicating the study.

Validity of the findings

Indeed, in terms of technologies and methods used, the novelty is not high, therefore the field of application and the results obtained justify the work. In fact, patterns have emerged within the data that help treat tuberculosis based on the general clinical picture.

The data has been provided.

The conclusions are well formulated and linked to the objective and based on the results obtained. I would appreciate it if they could go over any previously drafted research questions.

Additional comments

I am curious: few records were chosen for such a high number. Is this due to an overly stringent selection process? Does the "resistance" attribute have null values ​​which led to this large reduction in instances?
Have you tried expanding the records and seeing how the models perform?
Have you tried to analyze historical patient data, i.e. the patient's status changes over time, to evaluate resistance to the drug as the disease evolves?

Reviewer 2 ·

Basic reporting

The used english should be reviewed because in some parts there are missing subjects etc.
it is not clear how figure 2 is obtained, which results are showed and how it is possibile to extract the results from that images.
Considering that it seems more usefull to remove it
I am remarking to improve the clarity of the work because the explanation is more complex than the work itself

Experimental design

The experimental pipeline is not immediately clear, although the overall work is not complex.

Validity of the findings

They explain that from the logistic regression they obtained a probability distribution and using this distribution they estimate the average probability of developing drug resistance for each cluster.
Considering that we have three clusters, it seems from the images presented that the average probability found of acquiring resistance is not convincing. I think the model is saying the probability that a patient belongs to a ‘yes’ or ‘no’ class - considering drug resistance - instead of saying how likely it is that they will develop resistance.

Additional comments

I suggest to authors to add a figure reporting some examples of different choiche on clustering's hyperparameter

I think that after indicating Accuracy, ROC-AUC and F1score, it is possible to avoid the indication of recall because seems useless

---

## Round 0.2 · accepted · Accept

Based on the re-review and my own reading, your manuscript has been accepted for publication in PeerJ Computer Science.